# The Neuroprotective Effects of *Dendropanax morbifera* Water Extract on Scopolamine-Induced Memory Impairment in Mice

**DOI:** 10.3390/ijms242216444

**Published:** 2023-11-17

**Authors:** Sung Bae Kim, Hyun Yeoul Ryu, Woo Nam, So Min Lee, Mi Ran Jang, Youn Gil Kwak, Gyoo Il Kang, Kyung Seok Song, Jae Won Lee

**Affiliations:** 1Korea Conformity Laboratories, Incheon 21999, Republic of Korea; suaa10@kcl.re.kr (S.B.K.); rhyckato98@kcl.re.kr (H.Y.R.); sw89@kcl.re.kr (W.N.); somin14@kcl.re.kr (S.M.L.); brian@kcl.re.kr (G.I.K.); songks@kcl.re.kr (K.S.S.); 2Huons Foodience Co., Ltd., Geumsan-gun 32724, Republic of Korea; rose86mr@huonsfoodience.com (M.R.J.); kyg@huonsfoodience.com (Y.G.K.)

**Keywords:** *Dendropanax morbifera*, scopolamine, acetylcholinesterase, memory impairment, oxidative stress

## Abstract

This study investigated the neuroprotective effects of *Dendropanax morbifera* leaves and stems (DMLS) water extract on scopolamine (SCO)-induced memory impairment in mice. First, we conducted experiments to determine the protective effect of DMLS on neuronal cells. Treatment with DMLS showed a significant protective effect against neurotoxicity induced by Aβ(25–35) or H_2_O_2_. After confirming the neuroprotective effects of DMLS, we conducted animal studies. We administered DMLS orally at concentrations of 125, 250, and 375 mg/kg for 3 weeks. In the Y-maze test, SCO decreased spontaneous alternation, but treatment with DMLS or donepezil increased spontaneous alternation. In the Morris water-maze test, the SCO-treated group showed increased platform reach time and decreased swim time on the target platform. The passive avoidance task found that DMLS ingestion increased the recognition index in short-term memory. Furthermore, memory impairment induced by SCO reduced the ability to recognize novel objects. In the Novel Object Recognition test, recognition improved with DMLS or donepezil treatment. In the mouse brain, except for the cerebellum, acetylcholinesterase activity increased in the SCO group and decreased in the DMLS and donepezil groups. We measured catalase and malondialdehyde, which are indicators of antioxidant effectiveness, and found that oxidative stress increased with SCO but was mitigated by DMLS or donepezil treatment. Thus, our findings suggest that ingestion of DMLS restored memory impairment by protecting neuronal cells from Aβ(25–35) or H_2_O_2_-induced neurotoxicity, and by reducing oxidative stress.

## 1. Introduction

Neurodegenerative disorders are typically characterized by the accumulation of specific protein aggregates, but there are also cases that exhibit aggregations of multiple pathological proteins [1]. These diseases can be classified according to their primary clinical features, such as dementia, parkinsonism, or motor neuron disease, or according to the anatomic distribution of neurodegeneration, such as frontotemporal degenerations, extrapyramidal disorders, or spinocerebellar degenerations [2]. Scopolamine (SCO) is a muscarinic acetylcholine receptor antagonist that can induce cognitive dysfunction in animals and humans. This is similar to the cognitive decline observed in patients with Alzheimer’s disease (AD) [3]. Acetylcholine is a neurotransmitter that is widely present in the central nervous system and is involved in a variety of brain functions, including cortical development and activity, cerebral blood flow control, sleep–wake cycles, and cognitive and memory processes [4]. SCO was utilized in this study because the regulation of the cholinergic system is a key process in memory impairment. The main treatments for AD are donepezil (an acetylcholinesterase inhibitor) and NMDA receptor antagonists. These drugs can improve symptoms and pathologic features, but they can also have side effects such as nausea, vomiting, anorexia, and insomnia [5,6,7]. As a result, many patients are treated with conventional therapies and complementary and alternative (herbal) medicines. These therapies have been shown to be effective in several studies [8,9,10].

*Dendropanax morbifera* (*D. morbifera*) is an endemic and evergreen plant in southern Korea and its leaves, stems, and roots are widely used in folk medicine for the treatment of migraine headache, dysmenorrhea, and wind-dampness [11]. Previous studies also suggested that *D. morbifera* contains bioactive compounds such as polyphenolics, flavonol and polyacetylene, which are known for their neuroprotective effects [12,13,14,15]. *D. morbifera* extract was reported to ameliorate neuronal damage in an animal model of Parkinson’s disease by reducing neuroinflammation and to ameliorate D-galactose-induced memory deficits by decreasing inflammatory responses in the hippocampus [16,17]. In addition, *D. morbifera* also alleviated hippocampal impairment in cadmium- and mercury-induced neurotoxicity in rats and reversed the calcium-induced reduction in proliferating cells and differentiated neuroblasts in the hippocampus [18,19,20].

However, no studies have reported on the preventive potential of *D. morbifera* leaves and stems (DMLS) on SCO-induced memory deficits. Therefore, the present study investigated the effects of DMLS on SCO-induced memory deficits in mice.

## 2. Results

### 2.1. Analysis of DMLS

Using a newly optimized and validated method, a quantitative analysis of rutin, vitexin, syringin, chlorogenic acid, and neochlorogenic acid was performed on three different lots of DMLS extract. Each lot was measured in triplicate. The total rutin, syringin, chlorogenic acid, and neochlorogenic acid contents were expressed as mg per g of dry extract. Quantitative analysis of the DMLS showed that the final product contained rutin, vitexin, syringin, chlorogenic acid, and neochlorogenic acid (Figure 1).

### 2.2. Neuroprotection Assay

SH-SY5Y cells were simultaneously treated with Aβ(25–35) or hydrogen peroxide (H_2_O_2_) and DMLS to confirm the protective effect of DMLS on neuronal cells. When DMLS was treated at concentrations ranging from 2.34 to 300 µg/mL with Aβ(25–35) or H_2_O_2_, cell viability increased with statistical significance at DMLS concentrations of 9.38 µg/mL or higher (*p* < 0.05 or *p* < 0.01) (Figure 2A,B). Under different types of stress conditions, the higher the concentration of DMLS, the higher the protective effect on neuronal cells, indicating that DMLS has a protective effect on neuronal cell damage.

### 2.3. Effects DMLS on SCO-Induced Memory Impairment in the Morris Water-Maze Task

When the Morris water-maze task was separately analyzed for each of the 5 days of training, the negative group showed significantly prolonged escape latency on days 2 to 5 compared to the control group (*p* < 0.01). In contrast, the 125, 250, and 375 mg/kg DMLS and donepezil groups showed significantly shortened escape latency on day 5 compared to the negative group (*p* < 0.01, Figure 3A). In the probe test, the negative group spent significantly less time in the platform quadrant than the control group (*p* < 0.05). In contrast, the swimming time was extended in groups treated with more than 250 and 375 mg/kg DMLS and donepezil (*p* < 0.05, Figure 3B,C). Additionally, the locomotor activity was not statistically significantly different (Figure 3D). These results suggest that DMLS and donepezil can improve spatial memory in SCO-induced mice.

### 2.4. Effects DMLS on SCO-Induced Memory Impairment in the Y-Maze Task

The negative group showed significantly lower spontaneous alternation than the control group (*p* < 0.01). Spontaneous alternation was significantly elevated in the 125, 250, and 375 mg/kg DMLS and donepezil groups compared to the negative group (*p* < 0.01, Figure 4A). Additionally, there was no statistically significant difference in the total number of arm entries (Figure 4B).

### 2.5. Effects DMLS on SCO-Induced Memory Impairment in the Passive Avoidance Task

In the retention trials, the negative group showed a significant decrease in latency time compared to the control group (*p* < 0.01), while DMLS significantly prolonged latency time in a dose-dependent manner at all dose levels. Donepezil also increased latency time, with the optimal effect at 5 mg/kg (DMLS 125 mg/kg, *p* < 0.05; DMLS 250 mg/kg and 375 mg/kg, donepezil 5 mg/kg, *p* < 0.01; Figure 4B). There were no significant differences in latency time between groups in the acquisition trials.

### 2.6. Effects DMLS on SCO-Induced Memory Impairment in the Novel Object Recognition (NOR) Test

During the test period, mice in the control and DMLS-treated groups spent more time exploring new objects than familiar ones. In contrast, mice in the SCO-induced group spent a similar amount of time exploring new and familiar objects, and their Recognition Index (RI) was significantly decreased compared to that of the control group (*p* < 0.01). However, mice in the group treated with SCO and DMLS spent more time exploring new objects than familiar ones, and their RI was significantly increased compared to that of the SCO-induced group (*p* < 0.01, Figure 4D).

### 2.7. Effects DMLS on Parameters of Acetylcholinesterase (AChE) Activity and Acetylcholine (ACh) Contents

In the mouse brain excluding the cerebellum, AChE activity was significantly higher in the negative group than in the control group (*p* < 0.01). Treatment with all DMLS concentrations and donepezil significantly reduced AChE activity compared to the negative group (*p* < 0.01, Figure 5A). On the other hand, SCO treatment resulted in a decline in ACh levels in the negative group compared to the control group (*p* < 0.01). Treatment with all DMLS concentrations and donepezil preserved higher ACh contents than observed in the negative group (*p* < 0.01, Figure 5B).

### 2.8. Effects DMLS on Parameters of Oxidative Stress

The malondialdehyde (MDA) concentrations in the negative group were significantly higher than those in the control group (*p* < 0.01). Treatment with all DMLS groups and donepezil significantly reduced MDA concentrations compared to the negative group (*p* < 0.01, Figure 6A). On the other hand, SCO treatment resulted in a decline in catalase (CAT) activity in the negative group compared to the control group (*p* < 0.01). Treatment with all DMLS concentrations and donepezil preserved higher CAT activity than that observed in the negative group (*p* < 0.01, Figure 6B).

## 3. Discussion

Neurodegenerative diseases are characterized by the progressive deterioration of nerve cells, which can affect locomotion, language, perception, cognition, and memory, depending on the regions of the nervous system affected [21]. Medicinal plants play a significant role in the management of memory deficit. However, only a few natural sources have been extensively studied for their therapeutic properties in treating AD patients [22]. Dendropanax species have a long history of traditional use in the treatment of inflammatory and arthritic conditions [23]. Elsewhere, the leaves, stems, roots, and seeds of some species have been used as traditional remedies for lung problems, cough, and neurological disorders, including paralysis, stroke, and migraine [24,25].

In this study, we characterized the DMLS by identifying the major peaks obtained in reversed-phase HPLC analysis. Quantitative analysis of the DMLS revealed the presence of rutin, syringin, chlorogenic acid, and neochlorogenic acid. Syringin, rutin, and chlorogenic acid are all natural compounds that have been shown to improve brain function, memory, and learning. Syringin has been shown to improve learning and memory in a rat model of aging induced by hippocampal injection of quinolinic acid [26]. Rutin protects dopaminergic neurons from 6-hydroxydopamine-induced damage by significantly reducing oxidative stress and inflammation in the brain [27]. Chlorogenic acid enhanced autophagy by upregulating lysosomal function to protect SH-SY5Y cells from H_2_O_2_-induced damage in [28]. These compounds may be potential therapies for age-related cognitive decline and other brain diseases. However, the underlying preventive mechanism of DMLS extract in memory impairment is still unclear. Therefore, this study evaluated the effect of DMLS to potentially prevent and treat learning and memory deficits in an animal model of memory impairment.

First, experiments were conducted to determine the protective effect of DMLS on neuronal cells. Recent in vivo and in vitro studies have shown that the peptide Aβ(25–35) has neurotoxic properties similar to those of the full-length Aβ(1–42) [29,30]. H_2_O_2_ exposure can kill neural cells through apoptosis and necrosis. Oxidative stress is a key factor in the pathogenesis of neurodegenerative disorders [31,32,33,34,35,36,37,38]. Antioxidants have been shown to attenuate oxidative stress-induced neuronal cell damage [39]. In the present study, we found that DMLS pretreatment significantly protected neuronal cells against death induced by Aβ(25–35) or H_2_O_2_. This protective effect was likely due to the antioxidant and anti-inflammatory properties of DMLS. Previous studies have shown that DMLS protects neuronal cells against death induced by other neurotoxic agents, such as glutamate and β-amyloid. These findings suggest that DMLS may have a broad neuroprotective effect. However, more studies are needed to elucidate the precise mechanisms by which DMLS protects neuronal cells from death.

We evaluated the effects of DMLS by conducting the Y-maze, Morris water-maze, NOR, and passive avoidance tests. We also investigated the underlying mechanisms of these effects. Spatial short-term memory can be assessed using the Y-maze task [40]. In the Y-maze test, SCO decreased spontaneous alternation, whereas treatment with DMLS or donepezil increased spontaneous alternation. The Morris water-maze task is an effective tool for evaluating hippocampus-dependent spatial learning and reference memory [22]. In the Morris water-maze test, the SCO-treated group showed increased platform reach time and decreased swim time on the target platform, but this was alleviated by treatment with DMLS or donepezil. The passive avoidance task is a fear-motivated task used to assess learning and memory [41]. DMLS ingestion increased the recognition index in a concentration-dependent manner in the short-term memory assessment through the passive avoidance task. The NOR test is a commonly used behavioral assay for the investigation of various aspects of learning and memory in mice [42]. The memory impairment induced by SCO reduced the ability to recognize novel objects. In the NOR test, recognition improved with treatment with DMLS or donepezil. DMLS improved spatial short-term memory, spatial learning and reference memory, fear-motivated learning and memory, and recognition memory in mice with cognitive impairment induced by SCO. These results suggest that DMLS may improve cognitive impairment induced by SCO. DMLS appears to improve memory by improving hippocampal function and inhibiting acetylcholine breakdown. Therefore, we conducted additional studies to investigate whether DMLS improves hippocampal function and inhibits acetylcholine breakdown.

SCO targets the hippocampus and increases AChE activity in that organ [43]. In the mouse brain, except for the cerebellum, AChE activity increased in the SCO group and decreased in the DMLS and donepezil groups. As a result, ACh levels were higher in the DMLS and donepezil groups. Our research suggests that DMLS can improve hippocampal function and inhibit acetylcholine breakdown. This is because DMLS can directly target and inhibit AChE. Another possibility is that DMLS can indirectly affect AChE activity by increasing the production of other molecules that inhibit AChE. We need further research to support these findings.

We hypothesized that DMLS’s antioxidant and anti-inflammatory properties could contribute to its memory-enhancing effects. Superoxide dismutase (SOD), CAT, and glutathione (GSH) are important antioxidants that protect the body from ROS damage [44]. MDA is a byproduct of lipid peroxidation, which is a process by which ROS damage lipids in cell membranes. MDA levels are often used as a marker of oxidative stress [45]. We measured CAT and MDA, which are indicators of antioxidant effectiveness, and found that oxidative stress increased with SCO treatment but was mitigated by DMLS or donepezil treatment. These results suggest that SCO may induce oxidative stress, leading to memory impairment, and that DMLS may reduce oxidative stress, thereby improving memory impairment.

DMLS can help to remove free radicals (unstable molecules that can damage cells) and reduce inflammation, another factor that can damage cells. Additionally, DMLS has been shown to attenuate learning and memory impairment induced by SCO. This is likely due to its ability to increase acetylcholine levels in the brain. DMLS can also help to reduce oxidative stress and inflammation by increasing the levels of antioxidants in the brain, which can contribute to learning and memory impairment.

DMLS is a promising approach for the treatment of neurodegenerative diseases such as AD. However, there are some limitations to the current research, informing future research directions. DMLS research has been conducted only using in vitro and animal models. More research is needed to confirm the safety and efficacy of DMLS in humans. We plan to conduct a clinical trial to evaluate the safety and efficacy of DMLS in humans with neurodegenerative disorders.

## 4. Materials and Methods

### 4.1. Plant Preparation

#### 4.1.1. Plant Material and Preparation of DMLS Extract

*Dendropanax morbifera* was purchased from Hanna Arboretum (Goheung, Jeonnam, Republic of Korea). The voucher specimen (CNU 17002) has been deposited in the Herbarium of the College of Pharmacy, Chungnam National University. The product was manufactured in accordance with the principles of GMP (Good Manufacturing Practice). The dried aerial parts of *D. morbifera* (100 g) were soaked in 2 L water and extracted at 121 °C for 6 h. Then, the extracts were concentrated under reduced pressure and spray-dried to obtain the DMLS. This extract was stored at 4 °C until use.

#### 4.1.2. Manufacturing Process and Quantitative High-Performance Liquid Chromatography (HPLC) of DMLS

HPLC analysis was carried out using a liquid chromatographic system (Agilent 1260 series, Agilent, Santa Clara, CA, USA) equipped with a diode array detector (Agilent 1260 series). The standards were purchased from Sigma-Aldrich (St. Louis, MO, USA) and chromatographic separation was performed with a Phenomenex Luna C18 column (4.6 × 250 mm, 5 μm, St. Louis, MO, USA). The mobile phase was prepared from solvent A (aqueous 0.1% formic acid in HPLC water and acetonitrile) and solvent B (acetonitrile). The following gradient was used: A/B ratios of 95:5 (0 min), 85:15 (20 min), 80:20 (35 min), 60:40 (45 min), 95:5 (50 min). A flow rate of 1.0 mL/min was employed. The detection wavelength was recorded at 250 nm and the injection volume was 10.0 μL. Dried powder of material (1 g) was accurately weighed and placed into a 100 mL volumetric flask containing precisely 100 mL of water. Subsequently, it was extracted by sonication at 50 °C for 30 min. After cooling, the samples were finally filtered through a 0.45 μm membrane filter prior to injection into the HPLC system.

### 4.2. In Vitro Experiment

#### 4.2.1. Cell Line and Cell Culture

Human neuroblastoma cells, SH-SY5Y, were purchased from the Korea Cell Line Bank (Seoul, Republic of Korea). The cells were cultured with Dulbecco’s Modified Eagle’s Medium (DMEM, Sigma-Aldrich, Milwaukee, WI, USA) supplemented with 10% heat-inactivated fetal bovine serum (FBS, Life Technologies, Carlsbad, CA, USA), 2 mM l-glutamine, 100 units/mL penicillin, and 100 µg/mL streptomycin and incubated at 37 °C with 5% CO_2_.

#### 4.2.2. Neuroprotection Assay

In order to determine the neuronal cell protective effect of DMLS, neuronal damage was induced with two substances, Aβ(25–35) (amyloid beta, Sigma Aldrich Inc., St. Louis, MO, USA) and H_2_O_2_ (Samchun Chemicals, Seoul, Republic of Korea). SH-SY5Y cells were seeded in a 96-well plate at 5 × 10^4^ cells/well and cultured for 24 h. After removing the medium, DMLS at various concentrations was treated with 10 µM Aβ(25–35) or 300 µM H_2_O_2_ and cultured for 24 h. CCK-8 (10 μL) (CCK-8, Dojindo Molecular Technologies, Inc., Rockville, MD, USA) solution was added to 100 μL of cell culture medium per well, incubated for 2 h, and absorbance was measured at 450 nm using a microplate reader (SoftMax Pro5, Molecular devices, San Jose, CA, USA). The OD value of negative control was converted to 100% to determine the protective effect of the test substance against nerve cell damage. Samples were tested thrice.

### 4.3. In Vivo Experiment

#### 4.3.1. Experimental Animals

Eight-week-old C57BL/6N male mice were purchased from Orient Bio Inc. (Seongnam, Republic of Korea). Mice were housed under controlled conditions (temperature, 22.3 ± 0.5 °C; humidity, 45.2% ± 3.7%; 12 h light/dark cycle). Mice were kept under controlled conditions for the behavior test (temperature, 22.3 ± 0.5 °C; humidity, 45.2% ± 3.7%; illuminance, 100~200 lux). The room was also soundproofed to minimize noise disturbances. Mice were fed a laboratory rodent diet (Envigo, Indianapolis, IN, USA) and provided with water (treated with reverse osmosis) ad libitum. Animal care and experimental procedures were approved by the Institutional Animal Care and Use Committee of Korea Conformity Laboratories (IACUC number: IA21-00592).

#### 4.3.2. Mice Grouping and Treatment

Animals were distributed into six test groups (eight mice per group) based on their graded body weights, including a control group, a negative group, three DMLS dosing groups, and a positive group. The control and negative groups received distilled water as a vehicle orally. DMLS dissolved in the vehicle was administered to the mice at doses of 125, 250, and 375 mg/kg by gavage. Donepezil (Jubilant Generics Ltd., Mysore, India) was injected intraperitoneally as a positive control. All test agents were administered once a day for 3 weeks before initiation of behavioral tests. Behavioral tests were conducted in a blinded manner. During behavioral tests, all test agents were administered 1 h before each trial, except for the control group. Memory impairment was induced by 1 mg/kg of SCO (i.p.) (Sigma Aldrich Inc, St. Louis, MO, USA) 30 min after treatment with test agents.

### 4.4. Y-Maze Test

The Y-maze consisted of three white arms symmetrically connected at a 120° angle, each arm measuring 35 cm in length, 6 cm in width, and 13 cm in height. The maze floor and walls were made of white polyvinyl plastic. Mice were initially placed in one arm, and then the sequence and number of arm entries were monitored for 8 min. An entry was only counted when the mouse’s entire body, including its tail, had fully entered the arm. An actual alternation, defined as the combination of successive entries into each arm (e.g., ABC, BCA, or CAB, but not ACA), was counted, and the spontaneous alternation was calculated using the following equation: %alteration = [actual alteration/(total arm entries − 2)] × 100. The number of arm entries also served as an indicator for movement and locomotor activity.

### 4.5. Morris Water-Maze Test

The water maze was a circular tank (120 cm diameter and 50 cm height) with an escape platform (6 cm diameter and 29 cm height) centered in one of the four quadrants. The pool was divided into four quadrants (northwest, northeast, southeast, and southwest), each with a different visual cue (square, triangle, circle, and star) attached to the wall. The water was 22 ± 2 °C and made opaque with a nontoxic white paint. The hidden platform was 1 cm below the surface for place training.

Two days before the start of the trials, all animals were allowed to swim in the pool without a platform for 60 s each day to assess their locomotor activity and acclimatize them to the environment. Training sessions consisted of two trials per day, spaced 120 s apart, for five consecutive days. Mice were randomly released in one quadrant of the pool without the platform, and the escape latency, the time it took the mouse to reach the hidden platform, was measured. Mice were allowed 60 s to explore the platform. If a mouse failed to locate the platform on its own, it was gently guided to it and allowed to remain there for 15 s.

After the last day of training, a probe test was conducted in the pool without the platform. Animals were allowed to swim for 60 s in the quadrant opposite to where the platform had been located previously. During that time, the time spent in the target quadrant and the swimming distance were recorded. The behavior of the mice was tracked using a video tracking system (EthoVision XT, Noldus, Wageningen, The Netherlands).

### 4.6. Passive Avoidance Task

The passive avoidance task apparatus consisted of two connected compartments (30 × 30 × 30 cm) one illuminated and one dark, with electric rods on the floor of the dark compartment. In the acquisition trials, a mouse was placed in the illuminated compartment and the door to the dark compartment was opened after 10 s. If the mouse entered the dark compartment, the door closed automatically and the mouse received a 3-s electric shock of 0.3 mA. If the mouse did not enter the dark compartment within 90 s, it was gently guided in. In the retention trials, conducted 24 h later, the procedure was the same, except that the electric shock was not administered. The latency to enter the dark compartment was measured, up to a maximum of 300 s.

### 4.7. Novel Object Recognition Test

The apparatus used in this research was an open box (80 × 80 × 40 cm high) made of black acrylic. In the “sample” trial (T1), two identical objects were placed in opposite corners of the apparatus. A mouse was placed in the apparatus and allowed to explore the two objects for an unlimited amount of time. After T1, the mouse was placed back in its home cage for a 1-h inter-trial interval. The “choice” trial (T2) was then conducted. In T2, one of the objects from T1 was replaced with a new object. The mouse was thus exposed to two different objects: one familiar and one novel. Exploration was defined as directing the nose towards the object at no more than 2 cm and/or touching the object with the nose. The total time spent exploring the two identical objects in T1 and the time spent exploring the two different objects (familiar and novel) in T2 were recorded.

### 4.8. AChE Activity and Contents of ACh in Brain Tissues

After the last behavioral test, the mice were euthanized and their brains, except for the cerebellum, were removed. The tissue was homogenized in 1 mL of ice-cold phosphate-buffered saline (pH 7.4, Thermo Fisher Scientific Inc., Roskilde, Denmark) and centrifuged at 5000 rpm/min for 10 min at 4 °C. The supernatant was used to determine acetylcholinesterase (AChE) activity and acetylcholine (ACh) levels. After the protein assay, equal amounts of all samples were analyzed using the Micro Pyrogallol Red method. AChE and ACh assay kits (Abcam, Cambridge, UK) were used according to the manufacturer’s instructions.

### 4.9. Oxidative Stress Parameters

The CAT activity in erythrocytes was measured using the Beutler method [46]. This method involves the spectrophotometric monitoring of the decomposition of the substrate H_2_O_2_ at 240 nm. CAT activity (CAT, Abcam, Cambridge, UK) was expressed as units per gram of hemoglobin (U/g Hb). Lipid peroxidation levels in plasma samples were expressed as MDA (MDA, Abcam, Cambridge, UK). MDA levels were measured using the Ohkawa method [47], which involves the spectrophotometric monitoring of the reaction between MDA and thiobarbituric acid (TBA). MDA levels were expressed as nanomoles per milliliter (nmol/mL).

### 4.10. Statistical Analysis

All statistical analyses were performed using SPSS software 21 (SPSS Inc., Chicago, IL, USA). Data are presented as means ± SEM (n = 8). During the Morris water-maze task, we used repeated-measures analysis of variance (ANOVA) to compare groups (control vs. negative group or negative control vs. DMLS groups). For other tasks, we used Student’s *t*-test to test for statistical significance between the control and negative groups. We used one-way ANOVA to compare the negative control and DMLS groups to determine significant differences and equality of variance. A value of *p* < 0.05 was accepted as statistically significant.

## 5. Conclusions

This study evaluated the protective effects of DMLS against SCO-induced memory loss. We assessed SCO-induced memory and learning deficits, as well as increased oxidative stress and AChE activity, using behavioral studies including the Morris water-maze test, Y-maze test, and object recognition test. The study found that DMLS has neuroprotective effects against neuronal cells damaged by Aβ(25–35) or H_2_O_2_. In animal studies, DMLS was shown to attenuate learning and memory impairments induced by SCO. This was evident in the Y-maze, Morris water-maze, NOR, and passive avoidance tests. The study also found that DMLS treatment increased the activity of antioxidant enzymes such as CAT and reduced MDA, which are indicators of oxidative stress. This suggests that DMLS may protect against oxidative stress-induced neuronal cell damage. Overall, this study provides evidence that DMLS has potential as a therapeutic agent for the treatment of learning and memory impairment. However, more research is needed to confirm these findings in humans.

## Figures and Tables

**Figure 1 ijms-24-16444-f001:**
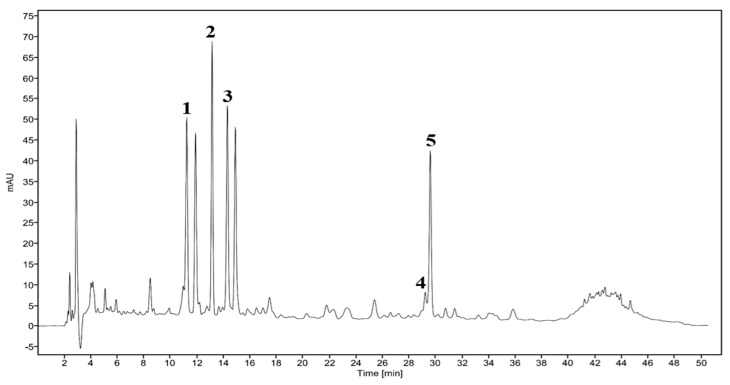
HPLC chromatogram of DMLS extract, detected at 250 nm: 1, neochlorogenic acid; 2, syringin; 3, chlorogenic acid; 4, vitexin; 5, rutin.

**Figure 2 ijms-24-16444-f002:**
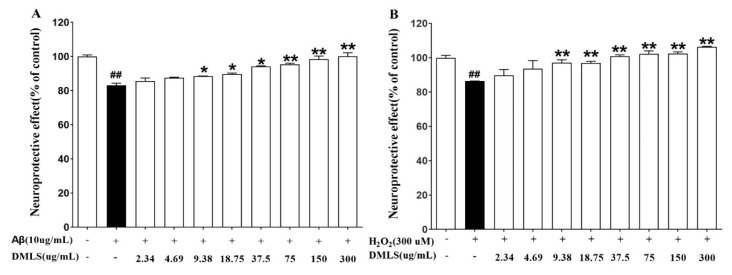
Protective effect of DMLS on neuronal cell damage induced by Aβ(25–35) and H_2_O_2_. SH-SY5Y cells were treated with (**A**) Aβ(25–35) (10 μM) or (**B**) H_2_O_2_ (300 μM) for 24 h. DMLS was then added at the indicated concentrations for an additional 24 h. Cell viability was determined by CCK-8 assay. Error bar: standard deviation; ##: significant difference compared with the negative control (Student’s *t*-test), *p* < 0.01; *: significant difference compared with the control, Aβ(25–35) or H_2_O_2_ (Student’s *t*-test), *p* < 0.05; **: significant difference compared with the control, Aβ(25–35) or H_2_O_2_ (Student’s *t*-test), *p* < 0.01.

**Figure 3 ijms-24-16444-f003:**
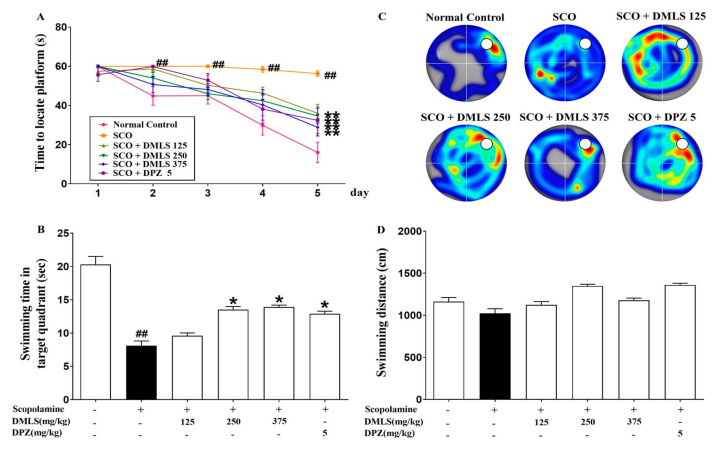
Effects of DMLS on SCO-induced memory impairment in the Morris water-maze task. To evaluate the effect of memory improvement, (**A**) the escape latency and probe test were measured in training sessions for 5 consecutive days. Mice were allowed to swim in the water tank without a platform for 60 s. (**B**) Time spent in the quadrant where the platform was previously located was recorded. (**C**) Representative heatmap images of probe trial sessions. Redder coloration indicates a longer time spent by mice in each quadrant. Conversely, a cooler color indicates a shorter duration. The white circles show the position where the platform was located previously. (**D**) The total swimming distance also was measured. Data are expressed as mean ± SEM (n = 8). ## *p* < 0.01 vs. normal control; * *p* < 0.05 and ** *p* < 0.01 vs. SCO group.

**Figure 4 ijms-24-16444-f004:**
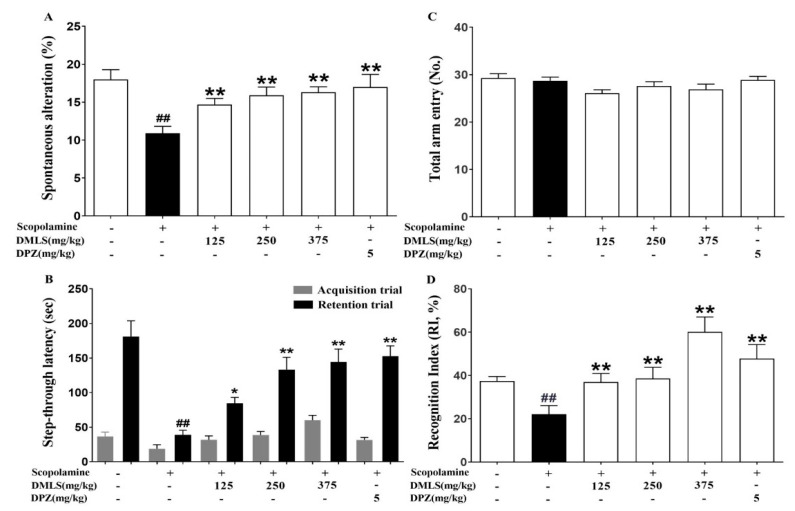
Effects of DMLS on SCO-induced memory impairment in mice. The Y-maze task is a test of spatial learning and memory. (**A**) Spontaneous alternation test. The number of spontaneous alternations in the Y-maze task. (**B**) Passive avoidance test. Latency time to enter the dark chamber in the retention trials. (**C**) Total arm entries in the Y-maze task. No significant difference was found between groups. The passive avoidance test is a test of long-term memory. (**D**) NOR test. The RI in the NOR test is a measure of the ability to discriminate between new and familiar objects. Data are expressed as mean ± SEM (n = 8). ## *p* < 0.01 vs. normal control; * *p* < 0.05 and ** *p* < 0.01 vs. SCO group.

**Figure 5 ijms-24-16444-f005:**
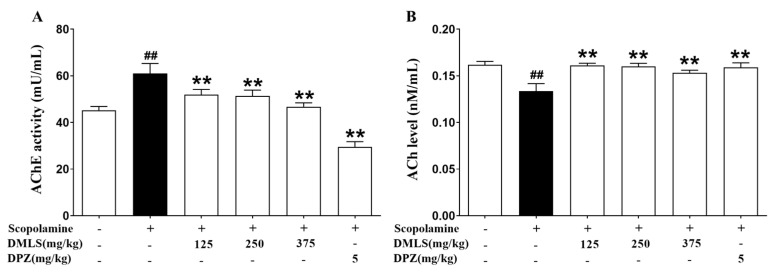
Effect of DMLS and donepezil on AChE activity and ACh contents in mouse brain. AChE is an enzyme that breaks down ACh, a neurotransmitter. (**A**) AChE activity in the mouse brain excluding the cerebellum and (**B**) ACh contents in the mouse brain excluding the cerebellum. Data are expressed as mean ± SEM (n = 8). ## *p* < 0.01 vs. normal control, ** *p* < 0.01 vs. SCO group.

**Figure 6 ijms-24-16444-f006:**
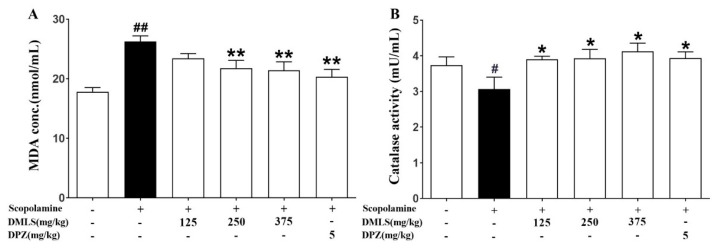
Effects of DMLS on MDA and CAT activities in rat hippocampus. MDA is a marker of oxidative stress. (**A**) MDA concentrations—the higher the MDA concentration, the more oxidative stress is present. CAT is an enzyme that helps to protect cells from oxidative damage. (**B**) CAT activity—the lower the CAT activity, the more oxidative damage is present. Data are expressed as mean ± SEM (n = 8). # *p* < 0.05 vs. normal control, ## *p* < 0.01 vs. normal control; * *p* < 0.05 and ** *p* < 0.01 vs. SCO group.

## Data Availability

All the data of the study can be made available upon request.

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
