# Peer review of "The Neuroprotective Effects of Dendropanax morbifera Water Extract on Scopolamine-Induced Memory Impairment in Mice"

_ijms, 2023, doi:10.3390/ijms242216444_

Round 1

Reviewer 1 Report

Comments and Suggestions for Authors

Dear Author, first of all, I would like to congratulate you for your work. However, there are a few shortcomings that need to be corrected and explained.

1. Latin names in the title and content of the article should be written in italics.

2. SH-SY5Y cell line is a neuroblastoma cell line. In the abstract and introduction sections of the study, you mention neurodegenerative diseases (such as dementia, parkinson, etc.). But why do you use a cancer cell line for its neuroprotective effect? Why didn't you use a neurodegenerative disease cell line? Or why did you choose the cancer cell line?

3. Please include limitations of the study.

Comments on the Quality of English Language

 Minor editing of English language required. 

Reviewer 2 Report

Comments and Suggestions for Authors the MS reported the neuroprotective effects of Dendropanax morbifera leaves and stems(DMLS) water extract on scopolamine-induced memory impairment in mice. DMLS contain lot of impurities. So, I strongly recommed the researchers to separate the mixture and perform the experiments. the crude mixture doesnt make any sense. So, I disagree with the conclusion of the study.  Comments on the Quality of English Language

the MS reported the neuroprotective effects of Dendropanax morbifera leaves and stems(DMLS) water extract on scopolamine-induced memory impairment in mice. DMLS contain lot of impurities. So, I strongly recommed the researchers to separate the mixture and perform the experiments. the crude mixture doesnt make any sense. So, I disagree with the conclusion of the study. 

Reviewer 3 Report

Comments and Suggestions for Authors

The article is devoted to the study of the neuroprotective effect of an aqueous extract of Dendropanax morbifera on scopolamine-induced memory impairment in mice. The article has high scientific and practical significance. This study will be of interest to specialists working in the field of studying neurodegenerative diseases. However, I have a few comments to improve the article:

1. In the discussion section, please add more information and conclusions about the results obtained in this particular article and the authors themselves, and not links and data from literary sources.

2. Please correct all figures, they have poor image quality and hard to read text.

Comments on the Quality of English Language

Minor editing of English language required.

Reviewer 4 Report

Comments and Suggestions for Authors

In this manuscript, the authors assessed the probable protective activity of the water extract of Dendropanax morbifera against the neurobehavioral and neurotoxic impacts in a murine model. Despite the importance of the topic, wide revisions are needed and several concerns are to be addressed as follows:

General comments:

1. There is a problem with using abbreviations throughout the manuscript. The full term should be mentioned first with the abbreviation between paresis then the abbreviations should be used throughout the manuscript. E.g., in the introduction, ROS should be presented as reactive oxygen species (ROS) at first mentioning then then the abbreviation should be used further. Also, Oxidative stress has been abbreviated as (OS) in the introduction then the full term has been repeated many times throughout the manuscript. Moreover, in the abstract: no need to give an abbreviation for recognition index (RI), Acetylcholinesterase (AChE), catalase (CAT), and malondialdehyde (MDA) as it has not been repeated in the abstract. Such errors have been repeated for most abbreviations throughout the manuscript.

2. Several grammatical, typing, and formatting errors exist (especially letter capitalization E.g. line 18 Passive Avoidance Task should be passive avoidance task; line 22 Acetylcholinesterase should be acetylcholinesterase…etc). Dendropanax morbifera should be italicized throughout the manuscript. Also, Dendropanax morbifera should be mentioned completely at its first mention then as D. morbifera further.

3. It is not preferable to begin sentences with an abbreviation like SCO in line 43 in the introduction.

Specific comment:

1. Abstract: more details about the invivo study should be added like the dose and route of DMLS and the duration of the experiment.

2. In the introduction the background on the bioactive and biological activities of Dendropanax morbifera should be more detailed.

3. The discussion needs to be deepened by more interpretations of the study findings and correlate the relation of the estimated parameters and the bioactive of  DMLS rather than just mentioning the results. Also, the authors should clarify the limitations of the study.

4. Material and the methods:

- The plant material should be identified by an experienced botanist to confirm the source and a voucher specimen should be deposited.

- The method of plant extraction needs to be illustrated in more detail. E.g. how the plant was dried? The reference to the method used is also missing.

- The authors have not clarified on what basis they have chosen the tested concentrations of all tested compounds including DMLS, scopolamine hydrobromide, and donepezil. Detailed justification of dose and route selection with references is highly needed. Also, the justification for the duration of the experiment should be explained. Moreover, justify the sample size of 8 mice per group.

- The behavioral test presentation lacks many important details. Has the observer been blind to the treatment? Describe the conditions of the test room in terms of temperature, light, soundproofing, etc. ). Also, clarify whether the tests were performed on a single day and at similar times.

- The ethical justification for the experiment and animal welfare details have not been clarified. What are the initial weights of the mice used?  How the mice were euthanatized? Has the mice were anesthetized?

- Complete information on kits used should be added as Trademark, city, and country of origin—also, detection range, sensitivity, inter and intra assay.

- Does data meet the assumption of homogeneity of variances and normal distribution? Clarify if the authors run a homogeneity or normality test.

- Why the authors have not examined the histopathological changes in the brain tissues?

Comments on the Quality of English Language

Several grammatical, typing, and formatting errors exist (especially letter capitalization E.g. line 18 Passive Avoidance Task should be passive avoidance task; line 22 Acetylcholinesterase should be acetylcholinesterase…etc). Dendropanax morbifera should be italicized throughout the manuscript. Also, Dendropanax morbifera should be mentioned completely at its first mention then as D. morbifera further.

Round 2

Reviewer 4 Report

Comments and Suggestions for Authors

The authors have satisfactorily addressed all questions raised by me. The manuscript is technically fine as of now.

Comments on the Quality of English Language

-